# Poly(carboxylated ether)s as Cement Additives: The Effect of the Addition Method on Hydration Kinetics

**DOI:** 10.3390/ma17215343

**Published:** 2024-10-31

**Authors:** Sara Beldarrain, Guido Goracci, Jorge S. Dolado, Aitor Barquero, Jose Ramon Leiza

**Affiliations:** 1POLYMAT, Kimika Aplikatua Saila, Kimika Fakultatea, University of the Basque Country, UPV/EHU, Joxe Mari Korta Zentroa, Tolosa Hiribidea 72, 20018 Donostia-San Sebastián, Spain; sara.beldarrain@ehu.eus; 2Centro de Física de Materiales (CSIC, UPV/EHU) Materials Physics Center (MPC), Paseo Manuel de Lardizabal 5, 20018 Donostia-San Sebastián, Spain; guido.goracci@ehu.eus (G.G.); j.dolado@ehu.eus (J.S.D.)

**Keywords:** hydration kinetics, PCE superplasticizer, direct addition, delayed addition, ordinary portland cement

## Abstract

Polycarboxylate ether (PCE) superplasticisers have been widely used in cement formulations. However, it is not until recently that several studies have analysed the relationship between the properties and the molecular structure. In the present work, PCEs with different side chain lengths and charge densities synthesised through free radical copolymerisation are used to analyse the effect they have on the hydration of ordinary Portland cement (OPC). It was found that the addition method of these PCEs to the OPC significantly affects the hydration kinetics of the cement paste. When PCEs are added through the direct addition method, a linear dependency between the retardation of hydration and the microstructure of the used PCEs is observed. On the contrary, when PCEs are added through the delayed addition method (PCEs are added to the cement paste 5 min after water), no retardation in hydration is observed, but the rate of hydration is reduced.

## 1. Introduction

It is reasonable to state that the advancement of polycarboxylate ether/ester (PCE) superplasticisers and concrete technology is happening concurrently [1]. PCEs are the newest class of superplasticisers that have replaced older chemistries based on lignosulfonates (LS), polynaphthalene sulfonates (PNS), or polymelamine sulfonates (PMS) [2] because of their exceptional capacity to reduce water. PCEs have comb-like structures, with anionic carboxylic groups forming the backbone and non-ionic polyethylene glycol (PEG) units acting as pendant groups (side chain) (Figure 1a). One of the advantages of these PCEs is their molecular versatility [3], as their microstructure can be easily varied by changing structural parameters like the length of the side chains [4,5], backbone length [6], the ratio between the carboxylic acid and side chains [7], etc. Due to their variety and capacity for reducing water, good workability [8] can be achieved at very low water-to-cement ratios. This makes PCEs indispensable in advanced concrete formulations, such as ultra-high-performance concrete (UHPC) [9] or self-compacting concrete (SCC) [10].

The knowledge of PCEs’ performance was based on a trial and error approach until recently, when Flatt et al., intending to understand the microstructure–property correlation, started to rationalise these relationships [11,12,13]. They defined a model repeating unit (Figure 1b), which describes the molecular structure of the PCE with only three characteristic structural parameters: the number of ethylene glycol units in the side chain (*P*), the number of monomer units in the repeating unit (*N*), and the number of repeating units in the chain (*n*).

In a later work, Marchon et al. [3] deeply analysed the effect that PCEs have on cement hydration. In their work, they examined the microstructure of PCEs synthesised by esterification with hydration retardation in specially designed model cement. In said work, the authors analysed the hydration kinetics using calorimetric measurements, examining two different methods for adding the PCE: direct and delayed addition. In direct addition, the PCEs were directly added to the water before mixing it with the cement to form the cement paste. In delayed addition, on the other hand, pure water was mixed with the cement first, and the PCEs were added five minutes after the cement paste was formed. They discovered that the hydration retardation caused by the superplasticisers was linearly proportional to the total number of carboxylate units dosed via the PCE, and this dependence was the same for both addition methods. The exception was that for direct addition, below a threshold PCE dosage (*c**), there was very little hydration retardation, and the relationship was not fulfilled. The fact that the observed retardation in hydration for both addition methods showed a similar dependence on the PCEs’ microstructures suggested that retardation was caused by the same mechanism in both cases. The authors obtained quantitative expressions that correlate the retardation in the maximum of the hydration peak (*Δt*) with the microstructures of the PCEs (defined as a function of the characteristic parameters) for both addition methods. Thus, Equations (1) and (2) rationalised the dependence of the hydration retardation on PCEs’ molecular structures, explaining this dependence as the result of the number of repeating units, $n_{RU}^{tot}$ or $\left(n_{RU}^{tot}-n_{RU}^{Ett}\right)$, and the blocking capacity of each of them, ${\left(\frac{C/E}{C/E+1}\right)}^{3/2}$, which depends on the electric field induced by the polymer on the surface.
(1)$$\mathrm{Delayed}\;\mathrm{addition}:\;\;\;∆t\propto n_{RU}^{tot}{\left(\frac{C/E}{C/E+1}\right)}^{3/2}$$
(2)$$\mathrm{Direct}\;\mathrm{addition}:\;\;\;∆t\propto \left(n_{RU}^{tot}-n_{RU}^{Ett}\right){\left(\frac{C/E}{C/E+1}\right)}^{3/2}$$
where *Δt* is the retardation of the maximum of the hydration measured using calorimetry (in hours), $n_{RU}^{tot}=\frac{c_{PCE}}{M_{RU}}$ is the number of repeat units dosed in the system, $n_{RU}^{Ett}$ is the number of repeat units adsorbed in ettringite, $c_{PCE}$ is the concentration of PCE (in µg_PCE_·g_OPC_^−1^), $M_{RU}$ is the molar mass of a repeating unit defined in Equation (3) (in g·mol^−1^), and C/E (or N − 1) is the molar ratio between carboxylate and ester groups in the PCE.
(3)$$M_{RU}=P\cdot M_{SC}+(C/E+1){\cdot M}_{BB}$$
where P is the side chain length, $M_{SC}$ is the molar mass of the side chain (ethylene oxide, 44.05 g·mol^−1^), (C/E+1) is the number of monomer units in the repeating unit, and $M_{BB}$ is the molar mass of the backbone monomer (methacrylic acid, 86.06 g·mol^−1^).

As mentioned, Marchon et al. carried out their research using PCEs synthesised through the esterification of a poly acrylic acid backbone with methyl poly(ethylene oxide), as they claim that this route offers the best control on the microstructure of the polymer. Nonetheless, there is an alternative synthesis method with wide use in industry, which is the free radical copolymerisation of methacrylic acid and polyethylene glycol methacrylate (PEGMA) macromonomer (see Figure 2). Emaldi et al. proved that when the chain length of the PEGMA was long enough to make it soluble in acidic media, copolymers with well-controlled molecular structures and homogeneous compositions could be achieved by following semi-batch polymerisation strategies [14,15,16]. This approach is the most common method due to its simple experimental process and cost-effectiveness [2].

The authors also studied the effect of PCEs produced by free radical copolymerisation on the hydration kinetics of commercial ordinary Portland cement (OPC). Their results show that the hydration kinetics of the PCE added using the direct addition method to commercial OPC follow the same trends described by Marchon et al. [3].

In our previous work, we expanded the study presented by Emaldi et al. with a wider range of PCEs, particularly focusing on having different compositions (varying N and P parameters) [17]. There, we observed a linear correlation between the hydration delay and the structure of the PCEs when the addition method of the PCE was direct addition. In the present work, a new set of experiments were carried out to investigate the effect of delaying the addition of the PCE to 5 min after mixing the cement and the water on the retardation of cement hydration. To ensure that the conditions were the same, the OPC and PCEs were from the same batch as those used in our previous work. We show that, surprisingly, the results of the delayed addition are significantly different from those of direct addition. Indeed, no hydration retardation is observed, but a reduction of hydration is observed.

## 2. Experimental Procedure

### 2.1. Materials

Two PCE series (M and L) were used with different ethylene oxide (EO) units (22.5 and 45) and containing different methacrylic acid/poly(ethylene glycol methacrylate) (MAA/PEGMA) mol ratios (0.67/1, 1/1, 3/1, and 6/1). The PCEs were synthesised with free radical semi-batch aqueous solution polymerisation in water, with KPS as the initiator at 80 °C. A detailed description of the synthesis and characterisation of these PCEs can be found in Appendix A. A summary of the most relevant characteristics of the synthesised PCEs is shown in Table 1.

The cement used was CEM type I 52.2R ordinary Portland cement (OPC), which was kindly supplied by Lemona Cements S.A. (Lemoa, Spain). Its mineralogical composition and specific surface area can be found in Appendix A.

### 2.2. Hydration Kinetics of the OPC

The hydration kinetics were carried out in a TAM air conduction calorimeter at room temperature for 48 h, with water as reference material. Cement pastes with a total mass of 5 g were prepared with a 0.4 water-to-cement ratio. The components were mixed using a vortex mixer. Different concentrations (between 1 and 4 mg_PCE_/g_OPC_) of the synthesised PCEs were added to the cement pastes following two different addition methods: direct and delayed addition.

In direct addition, the PCEs were first mixed with the water that later was added to the cement. As soon as the water and PCE mixture was added to the cement, the system was mixed as follows: 90 s at 800 rpm, 60 s of pause, and 90 s at 800 rpm. In delayed addition, the PCEs were added to the mixture 5 min after mixing the water and the cement. For this method, neat water was mixed with cement following the same mixing procedure described for direct addition. After 5 min, a PCE aqueous solution (at the concentration that was produced) was added, and the cement paste was mixed for an extra 60 s at 800 rpm. The same OPC without PCEs was used as a reference sample. The starting time of the measurement was taken as the moment when the OPC and the water were mixed.

The hydration kinetics were studied by measuring the released heat when the cement was mixed with water and PCEs. The added PCEs had different MAA/PEGMA ratios (N) and different side chain lengths (P) and were added at different dosages using direct and delayed addition.

## 3. Results and Discussion

### 3.1. Hydration Kinetics: Direct vs. Delayed Addition

In the left column, Figure 3 shows the hydration kinetics when the PCE is added using direct addition for the PCEs of series M (see Table 1) at different doses, and the hydration kinetics of delayed addition are shown on the right. The same results for series L can be found in Appendix A.

The results on the left side (direct addition) were adapted and reprinted with permission from our previous publication [17]. There, it can be observed that for any MAA/PEGMA ratio, as the PCE concentration increases, the hydration peak is more retarded. Notably, the overall shape of the plot is the same, but it is displaced to longer times. As it was discussed, the retardation in hydration (∆t) was found to be proportional to the carboxylate dosage, and all PCEs fit in a master curve, proving that the microstructure of PCEs could be correlated to the hydration delay of the OPC. These results are comparable to the delayed addition data published by Marchon et al. [3]. It is believed that this discrepancy is due to the content of C_3_A and the reduced role of ettringite in competing for the PCE adsorption [17,18].

The results on the right side of Figure 3 correspond to the experiments in which PCEs were added according to the delayed addition method. When this method is used, the overall shape of the plot significantly changes. Instead of only a retardation of the hydration, a decrease in the intensity of the main hydration peak is observed, but the position of the maximum of the peak is the same. This reduction of the peak becomes more significant when the MAA/PEGMA ratio or PCE concentration is increased. A retardation of the peak is only observed at the higher PCE doses. As a result of this behaviour, if only the position of the maximum is considered, the retardation (∆t) would be considered zero in most cases. Thus, the analysis of this behaviour cannot be conducted as a function of ∆t.

As mentioned, a reduction in hydration in delayed addition occurs during the first hours (<20 h) of hydration. Interestingly, the total released heat at 48 h (analysis time, see Figure 4) is comparable to that of the systems in which the PCEs were added using direct addition due to the appearance of a second peak at longer hydration times (>20 h). Because of this, we propose to base the analysis of the impact of these PCEs in the hydration kinetics of the OPC on the reduced heat of hydration of the main peak (compared to the reference), which is defined as $∆Q=Q_{ref}-Q_{sample}$. Thus, ∆Q will be used in the same way that ∆t was used in the pioneering work of Marchon et al. [3].

Figure 4 shows the heat of hydration (Q) over time for the PCE 3/1 M at different dosages using direct (continuous grey line) and delayed addition (dashed grey line) as an example. The rest of the systems show a similar trend and are presented in Appendix A.

It is noteworthy that the slope of the reference sample is similar to the slope of the direct addition samples, meaning that the rate of hydration is the same, even if a retardation is occurring. On the other hand, when PCEs were added through delayed addition, the hydration rate was slower (lower slope). Nevertheless, the total reduction of hydration is similar for both addition methods. Interestingly, the hydration rates of the direct and delayed systems become comparable when high PCE dosages are used.

### 3.2. Correlation of the Total Heat of Hydration with the Microstructure of the PCEs

During the last few years, several groups have been working on rationalising the effect of the microstructure of comb-shaped copolymers, such as the PCEs used in this work, in the properties of cement pastes [3,11,16,17,19]. As previously mentioned, Marchon et al. [3] were able to linearly correlate the microstructure and dosage of the PCEs used in a model clinker with its hydration retardation. Later, the equations used in that work for a delayed addition were validated in a commercial OPC for direct addition [17]. In the previous section, we could observe that when the same PCEs were added to the same OPC using a delayed addition, no retardation occurred. As this system shows a slower hydration rate, the released heat during the main hydration peak (corresponding to the combined C_3_S and C_3_A hydration) was analysed by applying the equations developed by Marchon et al. [3]

In Figure 5, the total released heat after 48 h is presented over the carboxylate dosage. This parameter, named *X_0_* is the number of carboxylate groups added to the system and can be calculated using Equation (4) and has a dependence on the PCE dosage ($c_{PCE}$), molar ratio of MAA/PEGMA (C/E), and molar mass of the repeating unit ($M_{RU},\;$ Equation (3)).
(4)$$X_{0}=c_{PCE}\cdot \frac{C/E}{M_{RU}}$$

The total released heat of hydration after 48 h plotted over the carboxylate dosage for PCEs from different series (M and L) and different addition methods (direct and delayed) showed that the total heat of hydration decreases when the carboxylate dosage increases. If both series are compared, a more negative slope is observed when M series are used. Independently of the series, the slope becomes more negative when delayed addition is used.

However, as ∆t was the analysed parameter in previous works, ∆Q was calculated for a more exhaustive study by considering the heat released during the hydration of C_3_S and C_3_A at the same time. ∆Q was calculated using Equation (5).
(5)$$∆Q=Q_{ref}-Q_{sample}$$
where ∆Q is the difference in the heat released during the main hydration peak between the reference ($Q_{ref}$) and each system ($Q_{sample}$). The main hydration peak was considered from point 1 to point 2, as shown in Figure 6.

The upper graphs shown in Figure 7 present the retardation of the maximum of the hydration (∆t) curve plotted as a function of the number of carboxylate groups (*X_0_*) calculated using Equation (4) for the direct addition method. In agreement with the results of Marchon et al. [3], the retardation of the hydration of the main peak increases linearly with the amount of carboxylate groups added to the system. In addition, for the same MAA/PEGMA molar ratio, the results of both series can be fitted to the same plot (as demonstrated in Figure 8 of our previous publication [17]).

On the other hand, the bottom graphs in Figure 7 show the reduction of the heat of the main hydration peak (∆Q) over the number of carboxylate groups (*X_0_*) added to the system for the delayed addition method. When series M is analysed, a linear dependence is observed, which is a function of the MAA/PEGMA molar ratio as in the case of ∆t in direct addition. Nevertheless, series L differs significantly from the results analysed so far. When PCEs with low MAA/PEGMA molar ratios are used, almost no reduction in hydration is observed, even if a difference was observed when analysing the hydration curves (see Figure 3 and Appendix A). Moreover, if the slopes for PCEs with the same MAA/PEGMA molar ratios and different series are compared, a significant difference is observed. Therefore, we can infer that ∆Q has a dependency on the side chain length (*P*). Because the system does not show any retardation in hydration and the heat released is *P*-dependent, the analysis proposed by Marchon et al. [3] and applied for a direct addition system when the same PCEs and OPCs were used [17] cannot be pursued further.

### 3.3. Understanding the Differences Between Direct and Delayed PCE Addition During Hydration

The lack of presence of PCE in the first 5 min of mixing cement with water is responsible for a significant change in the hydration curves due to a change in the hydration rate (see Figure 3 and slopes in Figure 4). This difference may be related to the change in surface area that occurred before the addition of the PCE in delayed addition. It is known that when C_3_S or C_3_A come into contact with water, pits are produced on the surface [20,21,22], resulting in an increase in the total surface area as it is displayed in Figure 8. This was confirmed in this work by measuring the BET surface area of the OPC as received and after blending the OPC and water for 30 min and quenching hydration and measuring the surface area again. The surface area increased by 35% after 30 min of hydration. Due to this increase, the same amount of PCE will have more area to cover in delayed addition. Therefore, the heat released during the hydration experiments under delayed addition do not show a delay, but they show a reduced hydration rate because not all the pits are blocked; hence, hydration occurs, but at a lower rate because there are less sites available for hydration. Due to the relatively short delay time (5 min), it is likely that the C_3_A phase was significantly more hydrated than the C_3_S phase; however, as the C_3_A content in the OPC used in this work is much lower than the C_3_S content (6.8% vs. 48%), the contribution of both mineral phases should be considered.

## 4. Conclusions

In this work, the effects of two different PCE addition methods (direct and delayed) were analysed and compared in terms of the hydration of a commercial OPC. As published previously, a linear dependency between the retardation of hydration and the microstructure of the used PCEs was observed when PCEs were added through a direct addition method. Conversely, when PCEs were added through a delayed addition method, no retardation in hydration was noticed. However, the rate of hydration was decreased, giving a reduction in hydration. Due to the reduced heat of hydration of the C_3_S peak, ∆Q was used in the same way that ∆t was used in the literature [3]. ∆Q was defined as the released heat difference of the first hydration peak between the samples with and without a PCE addition.

The reduction of the heat of the C_3_S peak was analysed over the carboxylate dosage, where different results were observed for both series. While the M series showed a linear dependence in all the MAA/PEGMA molar ratios, in the L series almost no reduction in hydration was seen, although a difference was observed in the hydration kinetics. Consequently, ∆Q cannot be used to rationalise the effects that PCEs with different microstructures have on hydration.

The fact that hydration is not affected by the addition mode of the PCE in a model cement [3], and on the contrary, the hydration is substantially affected in commercial OPC by the addition mode, suggests that the composition of the cements, in other words the crystalline phases, have different behaviours that are worth analysing in depth. This is ongoing research in our group, which will be disclosed in future publications.

## Figures and Tables

**Figure 1 materials-17-05343-f001:**
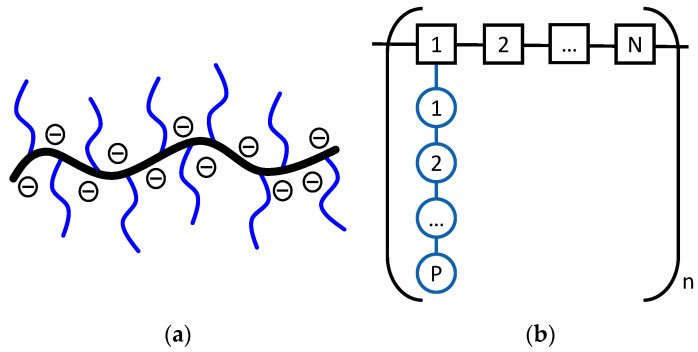
Schematic illustration of (**a**) typical PCE structure and (**b**) the repeating unit of a comb-shaped copolymer considered by Flatt et al. [11]. The comb copolymer contains *n* segments, each containing *N* backbone monomers and a side chain of *P* ethylene oxide units.

**Figure 2 materials-17-05343-f002:**
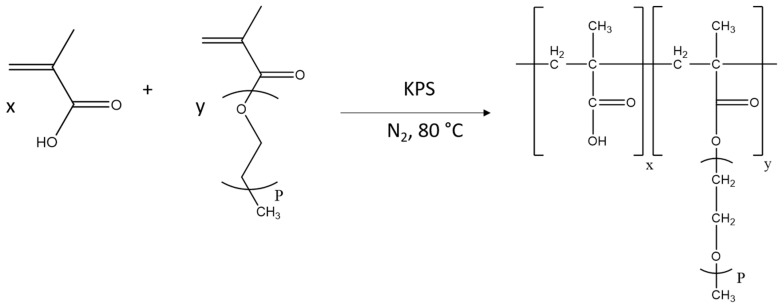
Mechanism for the free radical copolymerisation method to produce M-PEG-type PCEs.

**Figure 3 materials-17-05343-f003:**
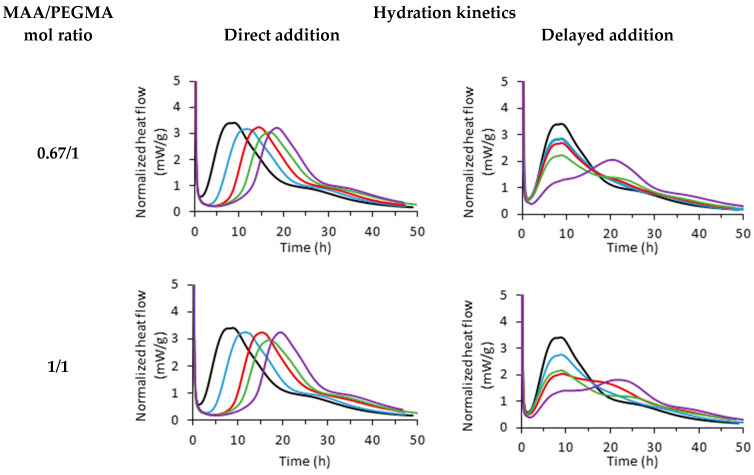
Released heat of hydration over time for PCEs of series M and different MAA/PEGMA molar ratios (0.67/1, 1/1, 3/1, and 6/1) at different PCE dosages (blue—1 mg_PCE_/g_OPC_; red—2 mg_PCE_/g_OPC_; green—3 mg_PCE_/g_OPC_; purple—4 mg_PCE_/g_OPC_). Black line belongs to the reference sample, to which no PCE was added. Data taken from reference [17].

**Figure 4 materials-17-05343-f004:**
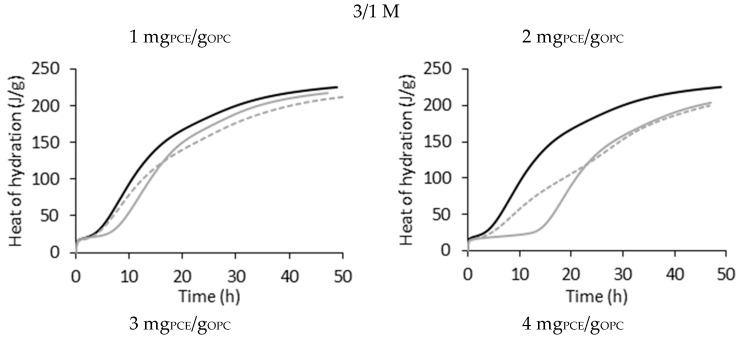
Total heat of hydration over time for the PCE of series M and an MMA/PEGMA molar ratio of 3/1 added to the system through direct (continuous grey line) and delayed addition (dashed grey line). Black line belongs to the reference sample, to which no PCE was added.

**Figure 5 materials-17-05343-f005:**
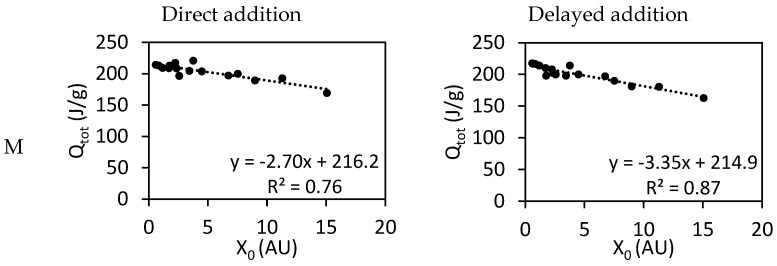
Total heat of hydration, $Q_{tot},$ as a function of the dosage of carboxylic groups, $x_{0},$ for cement pastes prepared through the direct and delayed addition of PCEs with different side chain lengths (M and L series).

**Figure 6 materials-17-05343-f006:**
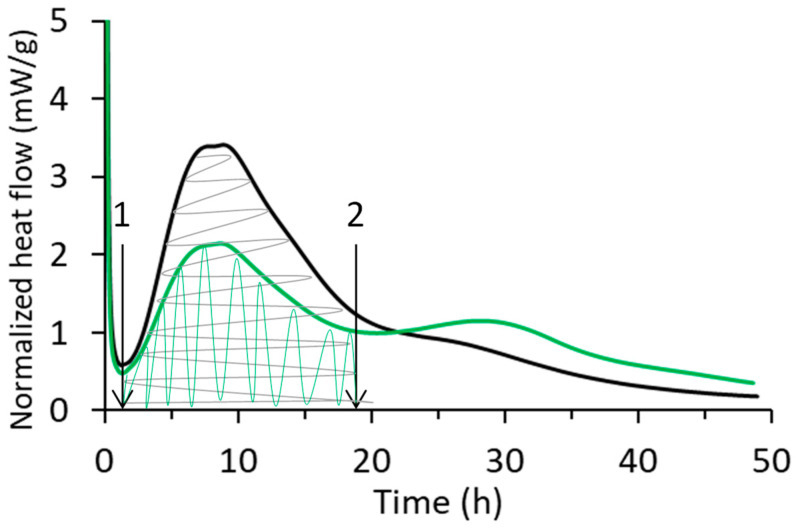
Representation of the areas used for ΔQ calculation considering the areas from the induction period (point 1) to the minimum or inflection point (point 2) for the reference and systems containing PCEs added using the delayed addition method. The black line represents the hydration calorimetry of the OPC with no PCE, and the green line the hydration calorimetry of the OPC with the PCE added by delayed addition method.

**Figure 7 materials-17-05343-f007:**
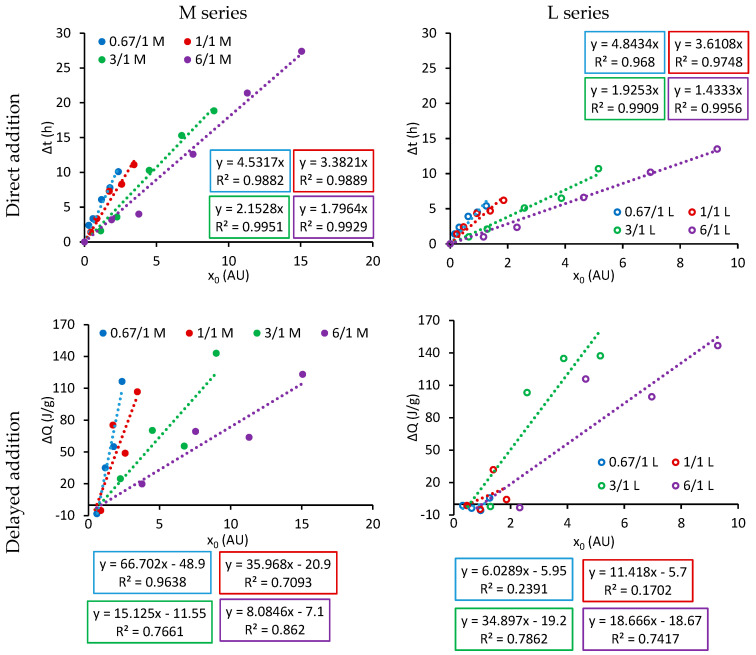
Retardation of hydration of the main peak in direct addition (**top** graphs) and reduction of hydration heat of the main peak in delayed addition (**bottom** graphs) as a function of dosage of carboxylic groups for PCEs of series M (**left** side) and L (**right** side) with different MAA/PEGMA mol ratios.

**Figure 8 materials-17-05343-f008:**
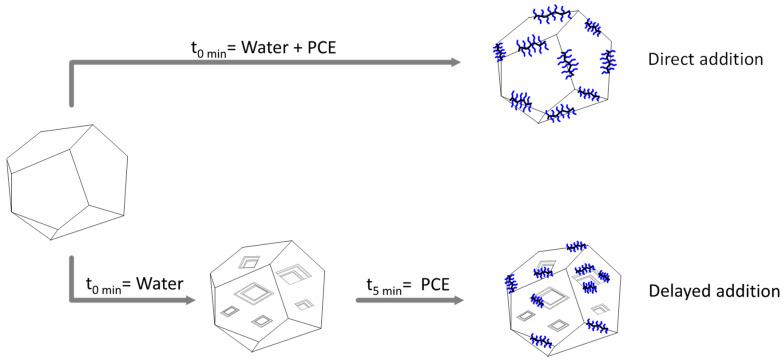
Schematic representation of a C_3_S crystal and its hydration when PCEs are present from the beginning of hydration (direct addition) and when PCEs were added after hydration starts (delayed addition).

**Table 1 materials-17-05343-t001:** Characteristic structural parameters of the used PCEs as defined by Flatt et al. [11].

Name of PCE	MMA/PEGMA Ratio (mol/mol)	$\bar{M_{n}}$ (kg/mol)	P	N	n
0.67/1 M	0.67/1	40.5	22.5	1.67	38
1/1 M	1/1	27.6		2	25
3/1 M	3/1	24.4		4	19
6/1 M	6/1	24.5		7	16
0.67/1 L	0.67/1	27.2	45	1.67	13
1/1 L	1/1	29.0		2	14
3/1 L	3/1	20.5		4	9
6/1 L	6/1	17.2		7	7

## Data Availability

The original contributions presented in this study are included in the article/Appendix A. Further inquiries can be directed to the corresponding authors.

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
