# Peer review of "Poly(carboxylated ether)s as Cement Additives: The Effect of the Addition Method on Hydration Kinetics"

_materials, 2024, doi:10.3390/ma17215343_

Round 1

Reviewer 1 Report

Comments and Suggestions for Authors

Comments on the Quality of English Language

none

Reviewer 2 Report

Comments and Suggestions for Authors

The paper "Poly(carboxylated ether)s as cement additives: The effect of the addition method on hydration kinetics" is interesting and provides a good insight into the issue of hyration kinetics of cement with  Polycarboxylate ethers as admixtures. The introduction is well done, and the research plan is clear and well prepared. 

There are however several issues I would like the authors to adderss: 

1. Description of the normalizing method for the start of the measurement would be beneficial to the paper. Seeing as there was a method of outside mixing used, there is quite a significant delay of start fo the measurement from adding water to cement, and for different samples it is different. This should be elaborated upon. I also suggest looking into internal mixing methods for TAM-air isothermal calorimetry if research into the delayed hydration is to be continued, as it may prove beneficial to the research process. 

2. How many tests were done for one sample type? 

3. The paper's discussion discusses only C3S as a cause for differences, what about C3A, which is much more active in the first several minutes of hydration? While it is not a wrong assumption, interaction between PCE and C3A was also discussed in many publications and maybe should also be explored. 

Minor issues: 

- figure number missing in line 232

Reviewer 3 Report

Comments and Suggestions for Authors

The paper is short but presents interesting finding in regards to effect of PCE with addition time. The authors are recommended to address the below comments for better clarity to the readers.

1.      Please correct the in-text reference citation format.

2.      Line 96 – The novelty part in the introduction is missing. Currently it is not clear, what is the knowledge gap or the reasoning behind the current study.

3.      Fig. 3 – It is interesting to see that with delayed addition, the PCE dosage doesn’t really retard the hydration. Why is that?

4.      And it also looks like there is a threshold dose of PCE (4 mgPCE/gOPC) beyond which the retardation takes place (as well as hydration reduction) with delayed addition. Does this threshold dose has any implication? Could the authors please elaborate on the possible reasoning?

5.      Line 225 – What is 7?

6.      Line 232 – Please provide figure number.

7.      Line 256 – Cartoon?

8.      Revise Figure 8 if possible to a better one.

Comments on the Quality of English Language

Minor editing of English required. 

Round 2

Reviewer 1 Report

Comments and Suggestions for Authors

author are suggested to revise the manuscript accroding to reviewers'comments. More explanations, statements,tests, references, discussions should be added. The response to reviewers is casual

Comments on the Quality of English Language

none

Author Response

Comment: author are suggested to revise the manuscript accroding to reviewers'comments. More explanations, statements,tests, references, discussions should be added. The response to reviewers is casual

Response: The reviewer is correct; some of the suggestions from the reviewers were not reflected in the first version of revised manuscript that was uploaded. This was an error from our side (we uploaded the wrong version of the document) and has already been corrected.

Reviewer 2 Report

Comments and Suggestions for Authors

The paper was improved in some part, however the discussion still is very limited in its scope, and the very addition of C3A leads to very unfortunate statement “main hydration peak (C3S and C3A hydration)" - the C3S and C3A have different peaks. Additionally, my comment was directed about the point 3.3. which was the mechanism discussion about the delayed addition of PCE - in the first 5 minutes C3S hydration is not all that developed while C3A reacts strongly at that time. I truly urge authors to comment on that, as I believe they current discussion is very lacking.

Author Response

Comment: The paper was improved in some part, however the discussion still is very limited in its scope, and the very addition of C3A leads to very unfortunate statement “main hydration peak (C3S and C3A hydration)" - the C3S and C3A have different peaks. Additionally, my comment was directed about the point 3.3. which was the mechanism discussion about the delayed addition of PCE - in the first 5 minutes C3S hydration is not all that developed while C3A reacts strongly at that time. I truly urge authors to comment on that, as I believe they current discussion is very lacking.

Response: We thank the reviewer for the clarification and apologize for not understanding their comment in the first round.

We are aware that the hydration peaks of C3S and C3A are separated, but for the purposes of this work, we have integrated the area corresponding to both of them at the same time. We have, nonetheless, reworded the sentences the reviewer is pointing out to avoid confusion of the reader.

We completely agree with the reviewer that the hydration of the C3A phase is faster than the one of C3S, and that we should have discussed that. We have modified the discussion on section 3.3 to reflect this. However, it is noteworthy that the C3A content of the OPC used in this work is about 6.8% while C3S is 48%, so we still believe that the C3S is the main responsible for the effects that we observe.

Round 3

Reviewer 1 Report

Comments and Suggestions for Authors

it can be accepted.

Comments on the Quality of English Language

none

Reviewer 2 Report

Comments and Suggestions for Authors

Authors have provided a sufficient explanation and made changes which improved the paper.